# A Novel Deep Learning-Based Classification Framework for COVID-19 Assisted with Weighted Average Ensemble Modeling

**DOI:** 10.3390/diagnostics13101806

**Published:** 2023-05-19

**Authors:** Gouri Shankar Chakraborty, Salil Batra, Aman Singh, Ghulam Muhammad, Vanessa Yelamos Torres, Makul Mahajan

**Affiliations:** 1Department of Computer Science and Engineering, Lovely Professional University, Phagwara 144411, Punjab, India; shankarchakraborty140@gmail.com (G.S.C.); salil.16836@lpu.co.in (S.B.); makul.14575@lpu.co.in (M.M.); 2Higher Polytechnic School, Universidad Europea del Atlántico, C/Isabel Torres 21, 39011 Santander, Spain; aman.singh@uneatlantico.es; 3Department of Engineering, Universidad Internacional Iberoamericana, Arecibo, PR 00613, USA; vanessa.yelamos@uneatlantico.es; 4Uttaranchal Institute of Technology, Uttaranchal University, Dehradun 248007, Uttarakhand, India; 5Department of Computer Engineering, College of Computer and Information Sciences, King Saud University, Riyadh 11543, Saudi Arabia; 6Engineering Research & Innovation Group, Universidad Europea del Atlántico, C/Isabel Torres 21, 39011 Santander, Spain; 7Department of Project Management, Universidad Internacional Iberoamericana, Campeche C.P. 24560, Mexico

**Keywords:** deep learning, convolutional neural network, image classification, COVID-19, ensemble prediction

## Abstract

COVID-19 is an infectious disease caused by the deadly virus SARS-CoV-2 that affects the lung of the patient. Different symptoms, including fever, muscle pain and respiratory syndrome, can be identified in COVID-19-affected patients. The disease needs to be diagnosed in a timely manner, otherwise the lung infection can turn into a severe form and the patient’s life may be in danger. In this work, an ensemble deep learning-based technique is proposed for COVID-19 detection that can classify the disease with high accuracy, efficiency, and reliability. A weighted average ensemble (WAE) prediction was performed by combining three CNN models, namely Xception, VGG19 and ResNet50V2, where 97.25% and 94.10% accuracy was achieved for binary and multiclass classification, respectively. To accurately detect the disease, different test methods have been proposed and developed, some of which are even being used in real-time situations. RT-PCR is one of the most successful COVID-19 detection methods, and is being used worldwide with high accuracy and sensitivity. However, complexity and time-consuming manual processes are limitations of this method. To make the detection process automated, researchers across the world have started to use deep learning to detect COVID-19 applied on medical imaging. Although most of the existing systems offer high accuracy, different limitations, including high variance, overfitting and generalization errors, can be found that can degrade the system performance. Some of the reasons behind those limitations are a lack of reliable data resources, missing preprocessing techniques, a lack of proper model selection, etc., which eventually create reliability issues. Reliability is an important factor for any healthcare system. Here, transfer learning with better preprocessing techniques applied on two benchmark datasets makes the work more reliable. The weighted average ensemble technique with hyperparameter tuning ensures better accuracy than using a randomly selected single CNN model.

## 1. Introduction

COVID-19 is considered as an infectious and contagious disease that can be transmitted from person to person [1]. It is mostly known for its deadly effects and high transmission rate. COVID-19 took the world’s attention when the WHO declared it as an epidemic [2]. The general symptoms of COVID-19 are quite the same those of as a normal viral fever patient, and it directly affects the respiratory system of the patient. So, at the early stage, it was a big challenge for healthcare systems to differentiate viral pneumonia-affected patients and COVID-19-affected patients. As the transmission rate was also very high, healthcare services were almost unable to control the situation, and were facing a lot of pressure to properly diagnose patients [3,4,5]. A detection system was required to detect COVID-19 patients efficiently so that they could be diagnosed in a timely manner.

To solve this issue, many researchers across the world came ahead and started working on developing such a detection method that can detect COVID-19. Different detection techniques have been introduced and suggested by different researchers. Reverse transcription polymerized chain reaction (RT-PCR) is widely being used and is considered one of the most successful methods for its accuracy and high sensitivity [6].

Despite having high efficiency and reliability, this particular method is quite complex, which can be observed from Figure 1. It also requires a lot of time to complete the procedure. Moreover, the process is lengthy and manual when the samples need to be collected and examined separately for each patient [7,8,9,10,11]. So, a smart and time-efficient system is expected, which would be simple to operate and could compete with existing manual methods. To make the process automated and quick, deep learning (DL) has come to the picture, where different DL-based models have been suggested and proposed by different researchers [12,13,14].

Deep learning (DL) is a subset of machine learning (ML) that is used in different smart real-time applications [15].

Figure 2 shows the basic steps of a deep learning system. Based on the architecture and applications, there are different types of deep learning techniques, such as CNN, RNN, autoencoders, GAN, etc. RNN is generally used for time series applications, whereas autoencoders are used for dimensionality reduction. CNN is widely popular for image classification-based applications, where it can classify the images by extracting the feature of the images [9]. GAN is used for generating synthetic images with an adversarial network form. To develop the image classification-based system, CNN is mostly used, where different researchers using CNN algorithms in their work [16,17,18,19,20,21,22]. One of the major facts of achieving improved accuracy is model selection. Although different predefined CNN models are already available to use, the best model needs to be selected to obtain a better outcome [6,8,23]. As COVID-19 is a different severe acute respiratory syndrome, it is very difficult to detect the coronavirus considering the physical symptoms. To overcome the problem, CNN models come with a solution where by extracting the features from the chest X-ray images, the disease can be identified by classifying the infected image [13].

The proposed work represents a deep learning-based COVID-19 detection approach that can classify COVID-19-affected lung images with improved efficiency, reliability and accuracy. Both binary and multiclass classification designs were developed, and the proposed approach can help to differentiate COVID-19 and viral pneumonia-affected lung images so that physicians can provide appropriate treatments to the respective patients. In summary, the contributions in this work are mentioned below:Three CNN models, including ResNet50V2, were trained with a large amount of medical images. Both binary and multiclass classification systems were developed with better preprocessing and classification techniques. Hyperparameter tuning ensures the models’ efficiency, and each model offers promising results.The weighted average ensemble technique was implemented, where pretrained models were combined and weights were assigned according to the priority of the models. The proposed ensembled technique ensures higher accuracy than when using a single model.Experimental evaluations were performed and presented to support the work with proper justification. The performance of the proposed system was compared with state-of-the-art methods, and the proposed work comes up with better results and improved performance.

The rest of the paper is arranged as follows: Section 2 presents the extensive related existing work, Section 3 highlights the problem definition, dataset descriptions, the microscopic description of the deep learning system and methodologies of the proposed COVID-19 detection system, and the functionality of the proposed system is presented. In Section 4, qualitative and quantitative analysis of the proposed work are thoroughly discussed with comparative analysis. In Section 5, the conclusion of the work is drawn.

## 2. Related Work

Today, deep learning is highly used in image processing applications such as image recognition with classification, feature extraction, synthetic image generation, etc. Deep learning is widely used in healthcare systems, especially in disease detection applications. To detect COVID-19, different papers with different techniques and systems have been proposed by different researchers, with deep learning (DL) techniques being applied in medical images, mostly in chest X-ray and CT scan images. Areej et al. [1] suggested a COVID-19 detection system based on binary classification that can identify loss with the loss function ‘binary cross entropy’. To train and test the model, 550 chest X-ray images were used, and 89.7% accuracy was achieved. Low accuracy may occur sometimes due to missing data preprocessing techniques. To improve the model accuracy, better preprocessing techniques should be applied.

Aparna et al. [2] designed a deep learning model that can classify infected images of COVID-19, where other pretrained models such as VGG-16 and MobileNet were compared by the authors with their proposed model. A web-based application was designed, through which physicians can diagnose their patients hassle-free. Only binary classification was performed where two datasets were used to train the model. Haritha et al. [4] developed a binary classification system using GoogleNet architecture, offering more than 90 percent accuracy. Although the model offers high accuracy, the number of data instances to train the model was much lower. The dataset, they used is consisting of 1824 images in total of both normal and COVID-19-affected patients. The system is only able to perform binary classification where only classification between two classes is possible. Tawsifur Rahman et al. [5] designed a system performing five-class classification, where DenseNet201 was used for two- and three-class classification tasks and InceptionV3 was used for five0class classification, having good accuracy. The dataset used in their work was a personal one created after observing 1937 patients.

Using image segmentation, infection in the lungs can be localized, and the severity level can also be measured by calculating the ROI of the infected area after localization so that physicians can provide timely diagnosis to their patients. Degerli et al. [6] performed a novel study by designing such an approach that can identify infection severity by generating an infection map. COVID-19-infected chest X-ray images can be classified in such a way that COVID-19 patients can easily be detected by identifying the infection status. The Qata dataset used in this work is considered as a benchmark dataset that has 120 k images, with respective mask images that help to achieve high accuracy and reliability. Heidari et al. [8] designed a COVID-19 detection system with blockchain and CNN, based on a privacy-aware method. Five different databases were used, with four classes (COVID-19, secondary TB, pneumonia and normal). Muhammad et al. [9] suggested a self-augmentation technique with a bidirectional LSTM and feature augmentation mechanism. The proposed techniques were nicely elaborated. In this work, both binary and multiclass classification have been performed, and a good comparative analysis has been shown. However, the data instances were low, which introduced reliability issues.

Nasiri et al. [11] developed an automated COVID-19 detection system with a deep neural network model, DenseNet169. It was used for feature extraction from chest X-ray images. Two datasets were utilized. The system offers high accuracy, so detection of COVID-19 can accurately be measured in an automated process. Basu et al. [12] applied a two-stage framework for COVID-19 detection, where feature extraction and feature selection algorithms were used. Although the proposed system offered good accuracy, the system may fail to identify early-stage infection from CT images. From the CT images, the CNN used in this work becomes unable to extract features, and that is considered a drawback of the system. Feature fusion is another technique, where features can be selected and extracted from one model and selected features can be used with another model, with the aim of obtaining better results.

Kong et al. [14] introduced the feature fusion technique with DenseNet and VGG16. A classification work was proposed where the feature fusion technique and better preprocessing techniques made the work successful with high accuracy. Two separate and publicly available datasets were used, consisting of 6518 medical images.

Gao et al. [16] proposed a system that can classify COVID-19 pneumonia-affected lung images so that doctors can provide treatments to their patients in time. Although the proposed model shows high accuracy, it has not been compared with other existing DL models used for COVID-19 classification work to justify the performances.

Sohan et al. [18] designed a CNN model having 22 layers, along with a detailed comparison with ResNet and VGG-16. Several datasets were used separately, with different accuracies found with different datasets. Combining multiple datasets may offer good accuracy and reliability with a higher number of instances rather than using the datasets separately. Table 1 represents the summary of the related work.

Akter et al. [20] implemented eleven pre-existing CNN models. MobileNetV2, VGG16, VGG19, InceptionV3, ResNet50, ResNet101 and NFNet were mostly highlighted in the work. A COVID-19 radiography dataset with chest X-ray images was used, where the system detects infected lungs through classifying the lung images. Narin et al. [21] suggested a system based on multiclass classification using the ResNet model. A very low number of instances were used to train the mode, which makes the reliability of the system questionable. However, the model can achieve higher accuracy.

Pham [22] proposed a system using CNN models to classify COVID-19, performing both binary and multiclass classification. AlexNet, GoogleNet and SqueezNet were the three pretrained models in the work. To train these models, data instances from six datasets were used. Almost 100% sensitivity was achieved with the SqueezNet model. No novel model was proposed in this work; only the pretrained models were used and discussed. Preprocessing techniques were not properly mentioned; a comparative analysis is missing. Jalali et al. [23] designed an evolutionary algorithm that can be used to detect COVID-19. Using the neighborhood labeling agreement, the accuracy of the proposed system improved. The KNN classifier replaced the softmax layer of the CNN model. From the basic version of swarm optimizer competitive algorithm, the evolutionary algorithm was designed and improved.

Ensemble techniques are used to improve the overall accuracy of a system. The ensemble prediction generally has higher accuracy than a single model’s prediction offers. Breve et al. [29] showed a comparison between ensemble techniques and single-CNN architecture performance. The main focus of the work was to take the most suitable techniques for detecting COVID-19. The COVIDx8B dataset was used, with around 16,000 images, to train models of different CNN architectures. After completion of the training, the models were tested and compared. The ensemble of CNN models was also employed. One drawback of this work was class imbalances, where the numbers of instances in the classes were different.

Srivastava et al. [30] designed an efficient DL algorithm that is novel and can detect COVID-19 by classifying chest X-ray images. The accuracy provided by the system is also good. A proper comparison of the performances of different CNN models was drawn, where each of the used model’s performances were mentioned and compared properly. No duplicate data were found, as the dataset was preprocessed properly and then split for training and testing. Moreover, each of the steps of the methodology was explained in detail. Shyni et al. [31] performed both binary and multiclass classification using a CNN model. Transfer learning was introduced, where different pretrained CNN models were used. A good and detailed comparison was given, with proper explanations and statistics. A limited number of CNN models were proposed for the multiclass classification task, and a good number of models were proposed for binary classification. Ahishali et al. [32] developed an advance warning methodology for a COVID-19 detection system. In this work, chest X-ray images were used. The developed system can generate warnings when the classified image is found to be affected by COVID-19. As a result, possible precautionary treatment can be provided to the COVID-19-affected patient before getting too late. Sumit et al. [33] developed a Yolov5 based classification technique for malaria detection using weighted ensemble technique. Neha et al. [34] proposed an image processing based COVID-19 detection system. Michael et al. [35] designed a COVID-19 classification system using transfer learning technique. Vasilis et al. [36] designed a simple and fast neural network for COVID-19 diagnosis using X-ray images. El-Kenawy et al. [24] proposed a COVID-19 classification approach in CT images using voting classifier and feature selection techniques. The guided whale optimization algorithm was used as a feature selection method, whereas the AlexNet model was used for the classification task. To implement the voting classifier, SVM, KNN and neural network-based classifiers were used. A total of 1128 CT images were applied during the training of the model.

Chang et al. [25] developed a deep residual network based on acoustics for COVID-19 diagnosis. Transfer learning was used where the ResNet50 model was trained. To enhance the recognition ability for the minority classes, a cost-sensitive technique and a data augmentation method were applied. Sen et al. [27] presented a novel study on COVID-19 prediction using an approach based on two-stage feature selection applied on chest X-ray images. The COVID-CT dataset was used, where 2482 CT images were utilized for the model training. Different guided feature selection techniques helped to optimize the final outcome, where 90% accuracy was achieved on average. Chang et al. [28] developed a unified approach using crowdsourced cough audio to detect COVID-19. The Dicova2021 dataset was used to train the ResNet50 model, where the achieved AUC_ROC was 85.43%.

Goel et al. [37] developed a novel optimized convolutional neural network-based approach for automatic diagnosis of COVID-19. To tune the hyperparameters, GWO (grey wolf optimizer) was used with the CNN models, where 97.78% accuracy was achieved. Song et al. [26] presented a study based on image segmentation for COVID-19 with a severity assessment. A DL method based on a self-supervised technique was used in this work. The work came up with high accuracy of 95.49%, with two public datasets and one private dataset being used to train the models. He et al. [38] designed a novel approach for developing an evolved adversarial network for the infection segmentation of COVID-19 with a gradient penalty. To accommodate the discrimination, an evolutionary population was composed with three generators. The proposed work offers high efficiency with novel features for COVID-19 infection segmentation. Shen et al. [39] proposed a detection algorithm for fog computing using deep features with discrete learning and PSO. The SVM classifier was used for classifying COVID-19.

## 3. Methodology

### 3.1. Problem Definition

Different testing methods are available to detect COVID-19, among which RT-PCR test offers high accuracy and sensitivity. However, because of having complex, time-consuming and manual procedures, an automated, efficient and smart detection system is required [7]. Deep learning plays an important role in terms of COVID-19 detection, and different deep learning (DL)-based systems have been suggested and developed by different researchers. Although those systems offer high accuracy, most of the systems are questionable in terms of reliability and performance in real-time uses [14]. Some of the common reasons that cause reliability issues are a lack of authentic datasets, better preprocessing techniques, proper model selection and a lack of comparative analysis and support.

To remove the reliability issue by considering these limitations, a proposed system has been developed that can perform both binary and multiclass classifications, offering high accuracy, efficiency and reliability. The system was developed based on CNN models, where the weighted ensemble technique ensures more accurate prediction. With the proposed system, not only can COVID-19 be identified, but pneumonia patients can also be differentiated from COVID-19-affected patients.

### 3.2. Dataset Description

To train deep learning models, data from reliable sources are required, because deep learning-based models work well with large amounts of data instances, which increases the reliability and efficiency of the performance. This section deals with the dataset information that has been used in the work to develop the proposed approach. Two datasets were selected to perform the classification task. The first dataset was the COVIDx CXR-3 (Covidx) dataset, which was utilized in particular for binary classification. To perform multiclass classification, the COVID-19 Radiography (CovidR) dataset was used.

Covidx dataset: COVID CXR-3 is an open-source initiative and is publicly available in Kaggle. It is considered as a benchmark dataset, with almost thirty thousand chest X-ray images available. The images are already preprocessed and classified into two different classes: ‘positive’ and ‘negative’. The only drawback of this dataset is the image size variation, with different sizes of images being available. Most of the images are of dimensions 1024 × 1024 or 513 × 460. Before feeding the instances to the models, images were resized to 200 × 200. 

From Figure 3, it can be observed that a total of 29,986 images were used for training, and 400 images were used for testing. In the ‘positive’ class, 15,994 images were there, and in the ‘negative’ class, 13,992 images were there. To remove the class imbalance, images of the ‘positive’ class were downsampled to 13,992.

COVID-19 Radiography dataset: CovidR is another benchmark dataset that is widely used for multiclass classification and infection localization. The dataset has 4 classes of images, including ‘Covid’, ‘Normal’, ‘Viral Pneumonia’ and ‘lung opacity’. The dataset also contains mask images for both lung and infection. To perform multiclass classification in this work, the ‘Normal’, ‘Viral pneumonia’ and ‘Covid’ classes were used. From Figure 4, it can be observed that 2500, 1345 and 3616 images are there in the ‘Normal’, ‘Viral pneumonia’ and ‘Covid’ classes, respectively. All the images used in this dataset were well preprocessed, having dimensions of 299 × 299. The images were reshaped and resized to 224 × 224 according to the input shape of the defined models before feeding. 

### 3.3. Deep Learning-Based Classification System

Deep learning (DL) is considered as a deep ANN (artificial neural network) that is a subset of ML (machine learning). Being a subset of ML, deep learning is widely used in different fields, including healthcare, business, industries, scientific research, etc. Deep learning techniques make the system accurate, reliable and automated; human interaction gets decreased, or there may not be any human interaction required at all [9].

DL is inspired by the function and structure of human brain. A basic neural network consists of three layers: input layers, one or more hidden layers and output layers. Input layers generally receive the input signal from user side. Once input data is received, it is sent to the hidden layers. Hidden layers are responsible for transforming and extracting the feature of the data to correctly classify it in output layer. Every layer is made of nodes or neurons. In a neuron, the actual computation is performed.

Deep learning models works well with large amounts of data, unlike machine learning models. However, the amount of training time taken by DL models is higher than the time ML models take.

To train a model efficiently, enough data instances are required from reliable data sources. There are different types of DL architecture, such as CNN, RNN, autoencoders, GAN (generative adversarial network), etc. CNN, or convolutional neural network, is generally used in image recognition and classification applications. RNN is used in time series applications, whereas autoencoders are used in dimensionality reduction and GAN focuses on generative modeling.

Figure 5 represents the steps of a basic classification model; to build a classification model, data instances need to prepared and preprocessed first. Then, data are split into two parts: one part for training and another for testing purposes. Then, model building and training is performed with the training data instances. Once the training is properly complete, the model performance can be evaluated using the test data through identifying output. The evaluation result gives an idea of how accurate the system is [3,4,6].

### 3.4. Microscopic Description of CNN Architecture

Convolutional neural network (CNN) is mainly used for image classification tasks. CNN models consist of sequences of layers, as shown in Figure 6, and can classify images after extracting high-level features from the images. It is a kind of feedforward network.

Some of the basic terms and parameters of CNN are as follows:

Convolutional layer: This is considered the main building block of a convolutional neural network, consisting of kernels or filters and some parameters that need to be learned during training [4].

Fully connected layer: Each node of one layer is directly connected with all nodes of the next layer and previous layer. However, no two nodes from the same layer are interconnected. Each of the nodes from a particular layer receives the complete information from the previous layers, then executes it and sends it to the next layer’s nodes for further processing [17].

Pooling: Pooling plays a significant role on feature extraction, where it pools high-level features from the image, such as edge and pixel data, to be processed. Max pooling, average pooling and sum pooling are the three types of pooling techniques that are mostly used [21].

Padding: Padding adds an extra row and column to the image pixel sample so that it can reduce the number of missing pixels during feature extraction.

Transfer learning: Transfer learning is defined as reusing a pretrained model with its existing knowledge and features so that a new result can achieved with better performance [13].

Hypertuning: To build and train a DL model, different parameters need to be considered. A set of hyperparameters are chosen for a learning algorithm, which need to be tuned so that the performance of the model can be improved. This helps to reduce the model’s errors by increasing the efficiency. Epoch number, learning rate, batch size, input size, etc., are examples of hyperparameters that need to be tuned during the training of the model so that efficiency can be improved.

Epoch: One single iteration to all the samples of training datasets. One epoch can have one or more than one batch.

Batch size: The number of samples that can be passed before updating the weights at a single time, which is considered by the model. If batch size is decreased, the number of steps in each epoch will be increased. In our work, the batch size was taken as equal to 32 for multiclass system, but 64 for binary classification system because of the higher number of instances.

### 3.5. Weighted Average Ensemble

WAE is defined as combining the trained models and assigning weights according to their performance to increase the overall accuracy of the system [29].

In Figure 7, prediction with weighted average ensemble technique is presented. Here, three models (M1, M2, M3) were trained separately, and later, their performances were combined after assigning some weights (W1, W2, W3) with values ranging from 0–1, respectively. The weights were assigned according to the priority of the models. The model among the three that offered best performance received the highest priority. The weight was assigned to each model though tensor dot operation with the model’s prediction value.

Three models (Xception (M1), VGG19 (M2) and ResNet50v2 (M3)) offered testing accuracy of 87.29%, 83.04% and 91.55%, respectively (shown in Table 2). According to the performance, ResNet50V2 had highest priority and VGG19 had lowest priority. So, ResNet50v2 had highest weight value (W3), Xception had medium weight value (W1) and VGG19 received lowest weight value (W2); therefore, W3 > W1 > W2.

Three weight values ranging from 0–1 need to be taken to maintain the priority order to check at which values the ensemble model performs better. Taking values manually is a lengthy and time-consuming procedure, and it is quite difficult to choose the three most suitable weight values for the three models. To make this task easily possible, grid search technique was used. Grid search is a hyperparameter tuning technique that can pick the most optimal value after trying out different values. The values 0.2, 0.1 and 0.4 for W1, W2 and W3, respectively, were selected to provide the best result.

### 3.6. Working Functionality of the System

Figure 8 represents the flow diagram of the proposed approach. At the starting phase, datasets of chest X-ray images are required. The images are preprocessed and split into two parts: training and test data. Training data are for training models and test data are used to evaluate the trained model performance. After training the three models, ensemble prediction is calculated. After assigning the required weights to each model, weighted average ensemble prediction can be performed. Then, test data can be used for the evaluation of the model by checking the performance to predict the accurate output.

### 3.7. Proposed Algorithm

The proposed Algorithm 1 has been designed for both binary and multiclass classification. The Algorithm 1 starts with fetching the instances from the dataset. Then data will be splitted and preprocessed properly. Once completion of the preprocessing, models will get trained one by one. After training completion, weighted average ensemble is done where weights are assigned to each models according to the performences. The Algorithm 1 ends once after calculating the weighted average ensembled accuracy.
**Algorithm 1:** Classification with Weighted Average EnsembleInput: Chest X-ray images from the datasetK: No. of epochM: number of modelsStart1.Get Dataset2. Extend Zipfile3.Split Data4.Preprocess data5.Set m to 06.Do Model Train7.      Set Model8.      Set epoch to 09.      While epoch < k + 110.   Get Preprocessed image data11.   Apply Data Augmentation12.   Pass to Model13.   Get Accuracy_score14.   Get Validation Accuracy15.   Repeat Steps16.  End while17.  Save Model18.  Load Model19.While m < 320.Set pred1, Pred2, Pred3, WEPred to 021.Pass input to Models22.Get score of Pred1, Pred2, Pred323.Set w1, w2 and w324.Set WEP = ([w1, w2, w3].[Pred,1 Pred2, Pred3])25.Get Weighted Average Ensemble Accuracy_scoreEnd

### 3.8. Experimental Evaluation

The system was implemented atop of TensorFlow, where several models were trained and evaluated. According to the performances on the respective datasets, three best models were selected for both the binary and multiclass classification systems represented by Table 2 and Table 3.

As the work is based on image classification, CNN-based DL models perform better than ML-based classifiers. It cannot be assumed which model performs better in which work. Proper model selection is important, rather than using all unnecessary models. Initially, five CNN models were selected for the classification task. After observing their performance, three best models were selected and the rest of the models were eliminated because of their poor performance. So, after training the three best models, they were ensembled with respective weights and achieved high accuracy.

Xception, VGG19 and ResNet50V2 were chosen for the development for both binary and multiclass classification-based systems, although Inception and VGG16 were primarily taken but later eliminated because of poor performances. The ResNet50V2 model offers better performance than other pretrained models with COVID-19 Radiography dataset and Covidx dataset for both binary and multiclass classification systems. For the final execution of the system, ResNet50V2 was selected for multiclass system and Covidx dataset-based binary classification system, and both of the systems offer reliable performance with their respective parameters. In this work, both binary and multiclass classification were performed. For binary classification task, in the final output layer, Sigmoid activation function was used, where Softmax was used for multiclass task. Sigmoid activation function receives a value from 0–1, with the threshold value of 0.5. Values that are less than 0.5 are assigned to 0; otherwise, 1. Softmax also takes a value ranging from 0–1, but it uses one-hot encoder to predict more than two classes’ values.

From Table 2 and Table 3, it can be observed that the weighted average ensemble design ensures higher accuracy for both the binary and multiclass classification systems. Although the performances of some other models are quite close, WAE accuracy provided more accurate results during testing of the system. In WAE, assigning weights depends on the priority of the models according to their performance and contribution. The model that shows better performance with maximum contribution receives the highest priority by being assigned the maximum weight. Lower weight is assigned to the models making smaller contribution for the system.

Different parameters are there, which need to be selected and set to obtain better performance. Without taking appropriate parameters, model’s performance can be degraded. This is why hyperparameter tuning plays important role in ensuring high performance. Different parameters need to be considered carefully, such as batch size, number of epochs, input image shape, learning rate, etc. Epoch number is one of the most important parameters that needs to be defined during model training. One epoch defines one single iteration to all the samples. The machine generally learns with increasing the number of epochs. However, sometimes, after a certain epoch numbers, accuracy may be decreased as the loss increases or there may not be any improvement in the accuracy. In those cases, call back and early stop functions can be an ideal solution that takes the particular number of epochs with higher accuracy and stops the iteration immediately if the accuracy starts to decrease. In our case, we took up to 12 epochs, where 8 epochs were found to be ideal for the proposed approach. Batch size defines the number of samples that can be passed before updating weights. Batch size was taken as 32. Input image shape was taken as 224 × 224, as the models gave better performance after being fed this particular size of image. Most of the images in the dataset had the same size, and rest of the images were resized with this shape before feeding. Learning algorithm ‘Adam’ was used, where learning rate of 0.001 was kept. After going through a long and systematic experiment, parameter values were selected accordingly.

In the proposed multiclass classification system, on the basis of the accuracy achieved by each three models, the priority was set for ResNet50V2, VGG19 and Xception, with 0.4, 0.2 and 0.1, respectively. Priority set for the binary system was ResNet50V2, VGG19 and Xception with 0.4, 0.4 and 0.1, respectively. Weighted prediction was determined first, with tensor dot operation between respective weights and prediction scores of each particular models. The model ResNet50V2 performed better in both binary and multiclass systems, where the accuracy scores obtained for both systems were more than 90%.

From Table 2 and Table 3, it can be easily observed that ResNet50V2 model performs well in both binary and multiclass classification systems with both of the datasets. ResNet, or residual network, is a very efficient convolutional neural network model that is highly known for solving the vanishing or exploding gradient problem. ResNet introduced the concept of residual blocks with skip connection technique, where skip connection permits connecting activation of a layer with further layer by skipping some of the in-between layers that are actually skipped. If the input is x, activation function is f(), bias is defined by b, weight is defined by w and output is O, then the output would be
O(x) = f(w × x + b) (1)
or
O(x) = f(x) where w = 1 and b = 0.(2)

However, in case of skip connection, the output would be
O(x) = f(x) + x

In ResNet, with the skipping technique in the residual block, the output would be
O(x) = Relu (f(x) + x)(3)

There are different versions of ResNet, such as ResNet50, ResNet50V2, ResNet101, ResNet101V2, ResNet152 and ResNet152V2. In our proposed system, ResNet50V2 is used. Figure 9 represents the ResNet50V2 architecture and Figure 10 presents the flow diagram of the ResNet50V2 model. Table 4 represents the summary of the ResNet50V2 model. The optimizer used in the system is Adam, which is a very decent and efficient one to provide a satisfactory result.

## 4. Result and Discussion

This section is divided into two subsections: Quantitative Analysis and Qualitative Analysis. This section provides a precise and concise description of the experimental results, their interpretation, and the experimental conclusions.

### 4.1. Quantitative Analysis

Quantitative metrics are generally used to check performance by calculating performance metrics [3]. Performance metrics such as accuracy, precision, recall and F1 score determine the performance of a system so that the performance can be evaluated and enhanced. Figure 11 represents the confusion matrix of the proposed binary classification technique. Table 5 represents the performance metrics of binary classification with the WAE technique.

Figure 12 represents the confusion matrix of the ResNet50V2 model-based multiclass system. Table 6 represents the performance metrics of the multiclass classification system with ResNet50V2.

Figure 13 represents the confusion matrix of the proposed multiclass system with the weighted average ensemble model. Table 7 represents different performance metrics of proposed and existing works.

Table 8 represents the comparative analysis between the proposed method with related existing work in terms of accuracy and the number of images and methods used. Figure 14 visualizes the comparison of the accuracy between the proposed work and existing work. The proposed binary classification technique with WAE (weighted average ensemble) offered 97.25% accuracy with red point, whereas multiclass classification achieved 94.10% accuracy with the WAE technique indicated by blue point. The bar graph representation in Figure 15 represents the performance metrics comparison of the proposed work with other existing work.

Both the static and k-fold cross-validation techniques were performed for splitting data into training and testing parts in the proposed binary and multiclass approaches. In the static method, train–test data were partitioned manually using the train_test_split module from the scikit-learn library, where different sizes of train–test data showed different performances, represented in Figure 16. According to the best performance, train–test data size was decided and the data were split accordingly. From Table 9, it can be observed that an 80:20 ratio of data splitting gave the best result for both the binary and multiclass-based approaches. In the multiclass approach, out of 7461 instances, 80% (5968) were taken for training and 20% (1493) were taken for testing. The testing data were again divided into two equal parts: test and validation. The validation data were used during the training of the model to reduce overfitting and generalization errors. The test data were used for final testing. The same approach was followed for binary classification, where 20% of the data (5597) from the training set (22,387) were also taken for validation. The test data were initially fixed as 400 for that particular dataset. Table 10 represents performance analysis with 80:20 ratio of data separation for 10 times experiments.

Table 11 represents the performance of the models used in the multiclass approach with the 5-fold cross-validation technique, where cross_val_score module from the ‘sklearn.model_selection’ library was used. The validation set was no longer required for this particular method. The dataset was divided into five smaller sets, or bins. Four sets were picked as the training set and 1 set was used as the test set. Five separate learning experiments were run, where 1 bin was picked each time as the test set and the rest of the others were kept as the training set. The mean accuracy was comparatively lower than the mean accuracy achieved from the static splitting method with an 80:20 ratio. Comparing the standard deviation values from Table 10 and Table 11, the static method provides a low scattered result, with 0.0117 and 0.0119 standard deviation for the multiclass and binary classifications respectively. Analyzing the performances from both static and cross-fold techniques, static splitting with an 80–20 train–test ratio was found to be more ideal for the proposed approach, although the static experiments took lot of time to determine the appropriate ratio. So, for the final evaluation, both the datasets were partitioned into an 80–20 ratio for training and testing.

### 4.2. Qualitative Analysis

Accuracy and loss graph for multiclass classification with COVID-19 Radiography dataset:

Figure 17 represents the graph of the train–test loss and accuracy of the ResNet50 model with 12 epochs for the COVID-19 Radiography dataset-based multiclass classification system. The number of epochs is represented with the *x*-axis, and the *Y*-axis represents the loss and accuracy rates. With a deep observation, it was found that initially, the training loss was very high (more than 0.4) and the accuracy was very low at the very first epoch. The loss was reduced and accuracy increased as the number of epochs increased, and after completing the fourth epoch, the recorded loss was less than 0.3 and the achieved accuracy was more than 88%. Keeping the epoch numbers increased, a better result was achieved with eight epochs, where the recorded loss was less than 0.25 and the availed accuracy was about 92%. So, as the epoch numbers got increased, accuracy got increased and loss got decreased. Continuing the process by increasing the epoch number to 12 epochs, it was observed that loss started to increase, with a recorded loss of more than 0.25, and accuracy decreased. So, for the proposed multiclass classification system, eight epochs were fixed as an ideal number to train.

Figure 18 shows the performance of the three models used in the multiclass classification system. Figure 19 represents the accuracy and training validation loss of the ResNet50V2 model used in the binary classification system with the Covidx dataset. From this graph, it can be seen that training and validation accuracy increased and loss decreased with the increase in the number of epochs. The training time taken by the system for each epoch was very high, as the number of the training instances was very large. So, if we go through the graph carefully, the accuracy line horizontally moved forward without notable fluctuations after completing the first epoch. So, keeping the epoch numbers increased, the accuracy hardly changed. Four epochs were taken, and the same result was found as shown in the graph.

## 5. Conclusions

COVID-19 has become a common disease now, and people are more aware of its fatal effects. However, fluctuations in the daily case rate can still be observed, and new cases are being found, but the infection rate and death rate have become low. This has been possible because of timely diagnosis. To ensure timely diagnosis, the disease needs to be identified at an early stage, and COVID-19 detection techniques are playing a significant role in identifying COVID-19 so that precautions can be taken to reduce the risk. Several test methods have been suggested, developed and are being developed. RTPCR is known as one of the best test methods and is widely used with a high success rate. In this testing method, samples need to be collected and examined manually. Because it is complex and time-consuming, this particular technique needs to be replaced by an automated one, where deep learning techniques can be used to detect infection in lung images. Most of the existing and conventional deep learning-based systems fail to provide desired results due to a lack of efficiency and reliability issues.

In this work, deep learning-based classification approaches were introduced, where both the binary and multiclass classifications can be possible with higher accuracy and efficiency. In the future, improved physical devices for COVID-19 detection can be developed with the proposed approach, with additional features and technologies. A classification system for detecting the variants of COVID-19 may also be possible to develop the use of advanced feature selection techniques with the hybrid usage of CNN and non-CNN models. Different optimization techniques, such as the genetic algorithm or swarm intelligence-based metaheuristic techniques, can be used; various optimization algorithms, such as ant colony optimization (ACO), artificial bee colony optimization (ABC), fish swarm optimization (FSO) or dragonfly optimization, can be applied for feature selection and extraction. Various machine learning classifiers such as support vector machine, K-nearest neighbor (KNN), decision tree (DT), logistic regression (LR), naïve Bayes (NB) and random forest classifiers can also be used for feature selection. Voting classifiers can be used to ensemble the machine learning classifiers, which could be a better heterogeneous feature selection technique. Once the appropriate features are selected, both homogeneous and heterogeneous ensemble techniques can be applied to CNN models to classify the results. To improve data preprocessing, autoencoders can be used to reduce extra dimensions and noise in raw data. GAN (generative adversarial network) algorithms can be used for data augmentation by generating large amounts of synthetic data, which can add an extra edge to improving the overall performance. However, only detecting the disease is not sufficient to provide enough help to physicians for timely diagnosis. It will be more helpful if the infection can be localized in the lungs and the status of the infection can be identified, so that severity measurement would be possible and physicians would be able to provide treatment based on the severity level. With the same purpose, our future work is to design a deep learning-based COVID-19 infection localization system in which infected areas in the lung can be visually identified by measuring the infection ratio. The proposed work could play an effective role in COVID-19 diagnosis, where doctors would be able to provide rapid treatment to their patients. Researchers and those who are interested in the same area could also find this work valuable for their own research.

## Figures and Tables

**Figure 1 diagnostics-13-01806-f001:**
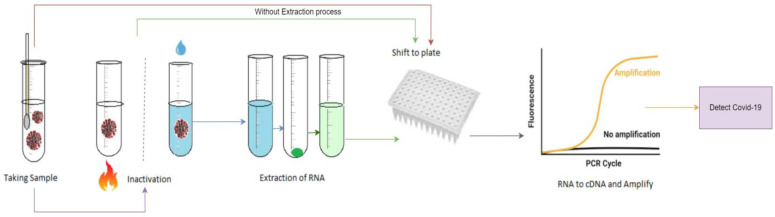
General steps of RT-PCR test method.

**Figure 2 diagnostics-13-01806-f002:**
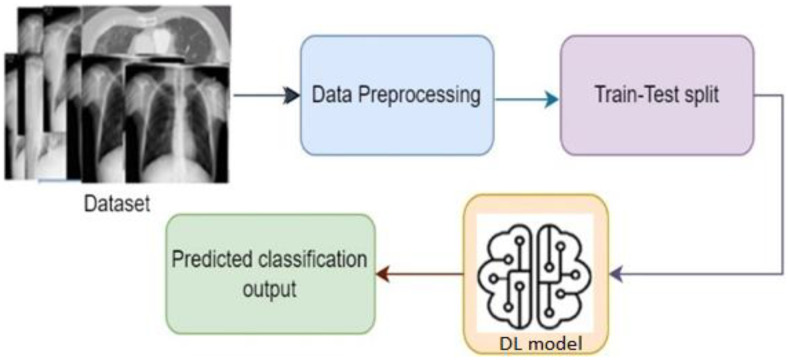
Basic steps of a DL-based system.

**Figure 3 diagnostics-13-01806-f003:**
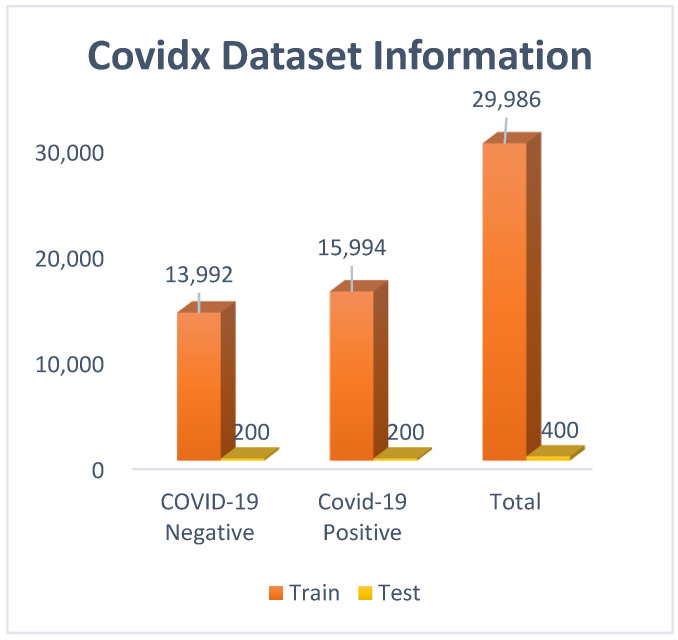
Covidx (Covid CXR-3) dataset visualization with bar graph.

**Figure 4 diagnostics-13-01806-f004:**
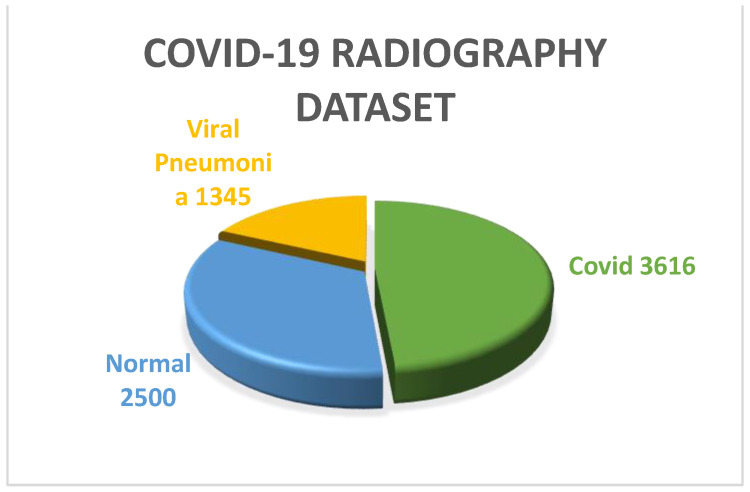
Pie chart representation of COVID-19 Radiography dataset.

**Figure 5 diagnostics-13-01806-f005:**
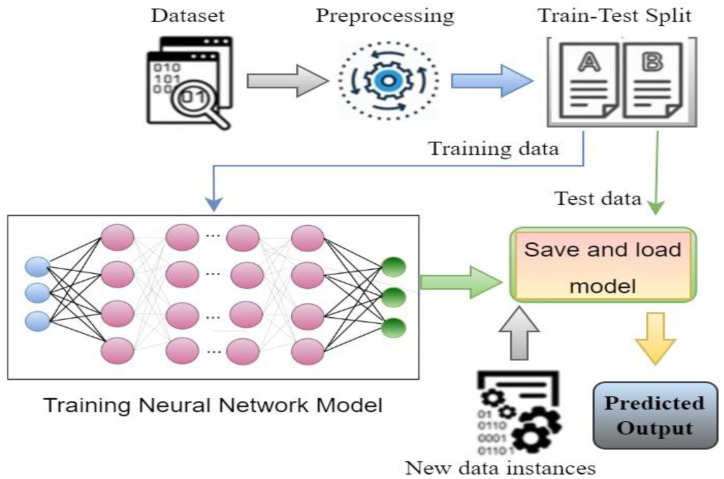
Steps to build a classification model.

**Figure 6 diagnostics-13-01806-f006:**
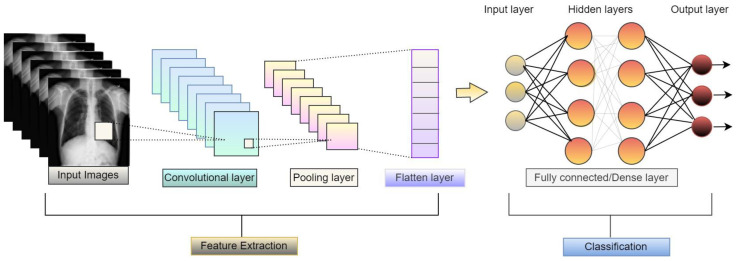
Basic architecture of convolutional neural network.

**Figure 7 diagnostics-13-01806-f007:**
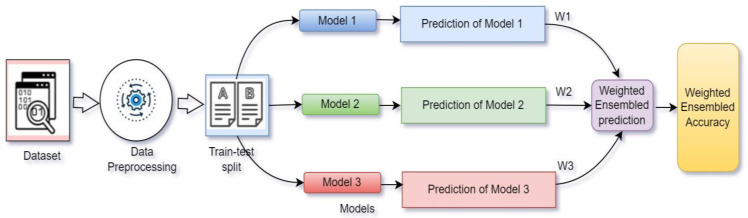
Weighted average ensemble technique.

**Figure 8 diagnostics-13-01806-f008:**
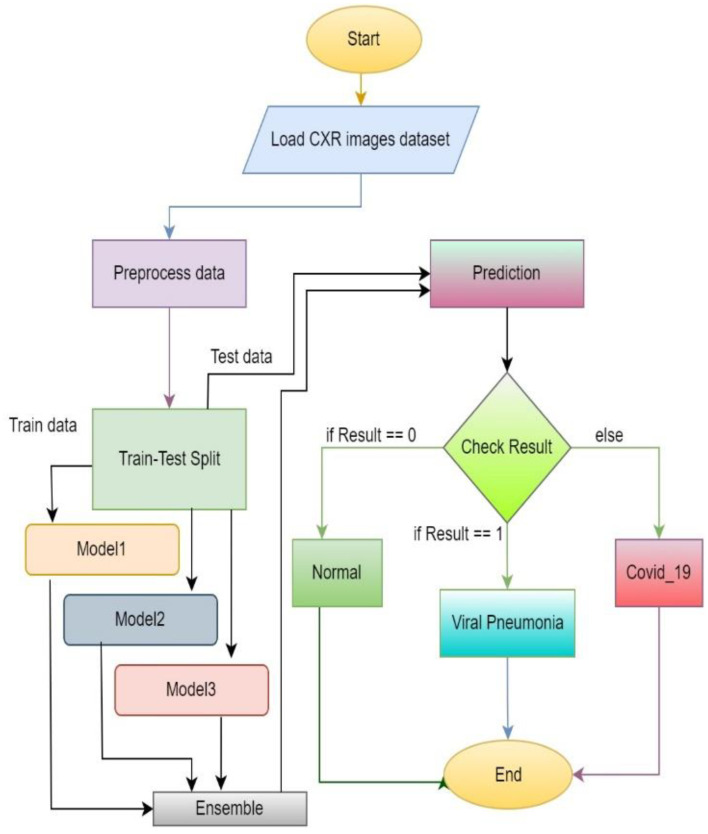
Flow diagram of the proposed system.

**Figure 9 diagnostics-13-01806-f009:**
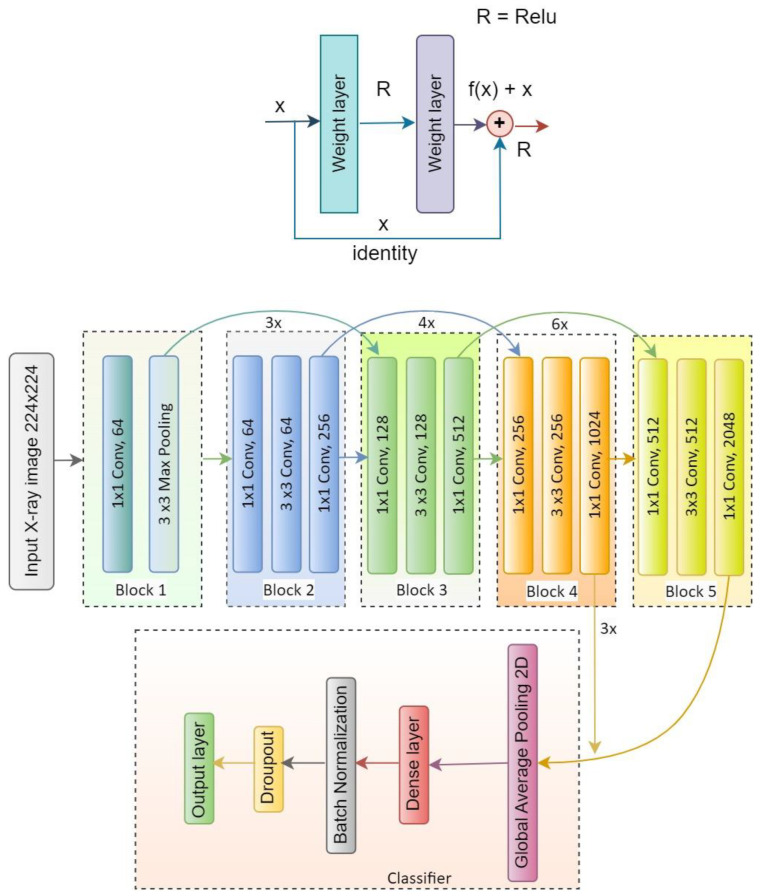
Architecture of the ResNet50V2 model.

**Figure 10 diagnostics-13-01806-f010:**
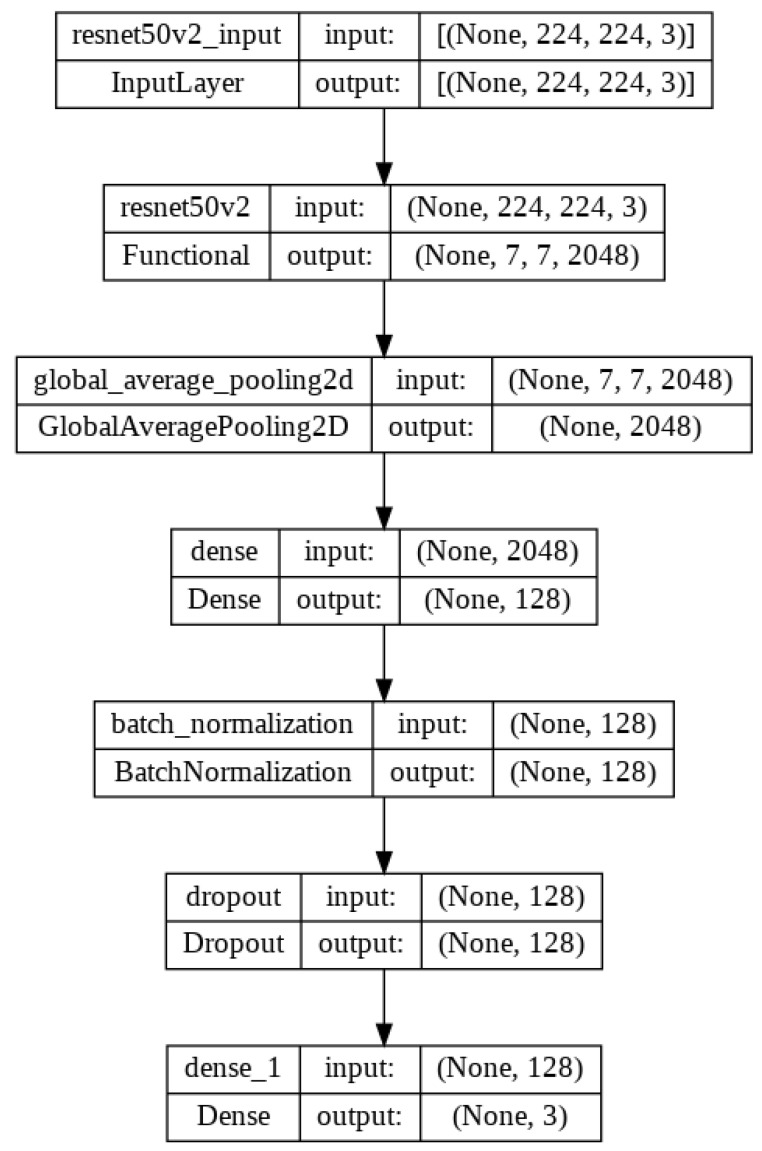
Flow diagram of the proposed model architecture with ResNet50V2.

**Figure 11 diagnostics-13-01806-f011:**
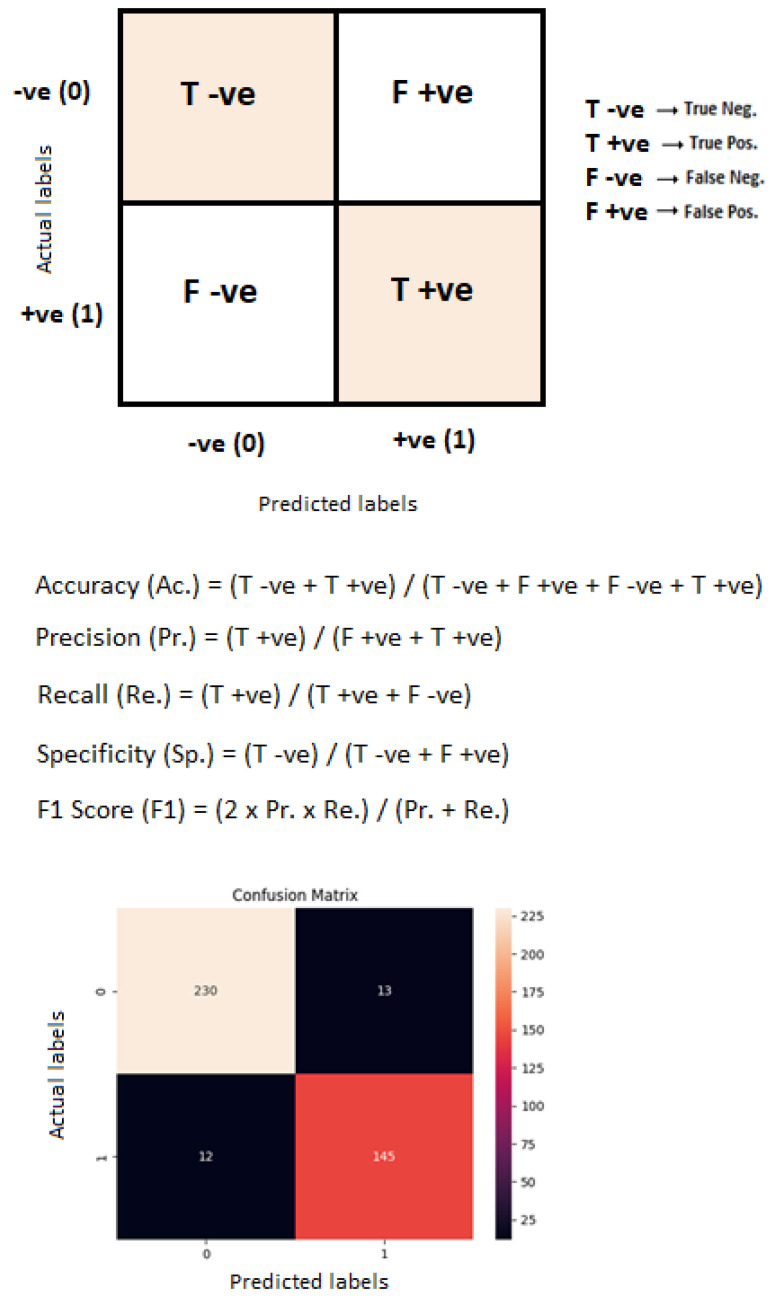
Confusion matrix of the proposed binary classification technique.

**Figure 12 diagnostics-13-01806-f012:**
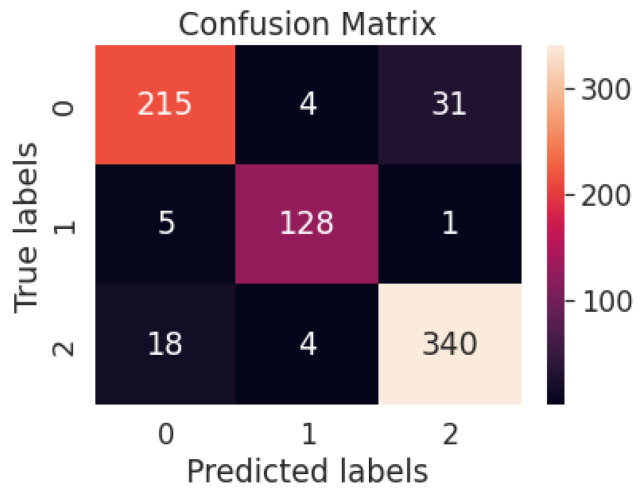
Confusion matrix of the ResNet50V2 model-based multiclass system.

**Figure 13 diagnostics-13-01806-f013:**
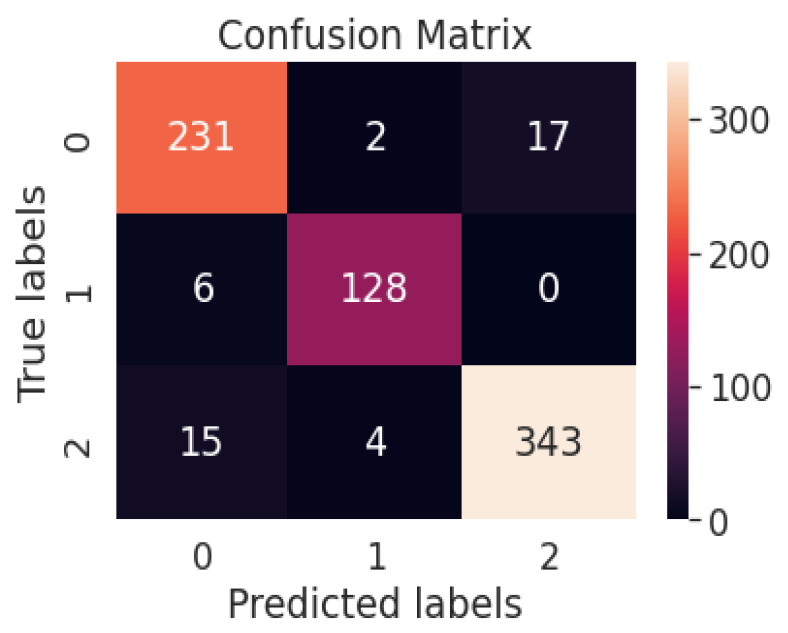
Confusion matrix of the proposed multiclass system with weighted average ensemble model.

**Figure 14 diagnostics-13-01806-f014:**
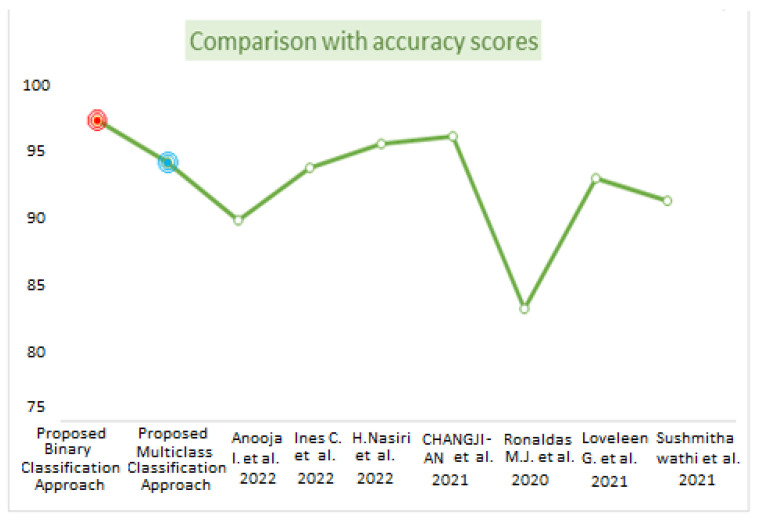
Comparison based on accuracy scores [3,10,11,13,15,17,19].

**Figure 15 diagnostics-13-01806-f015:**
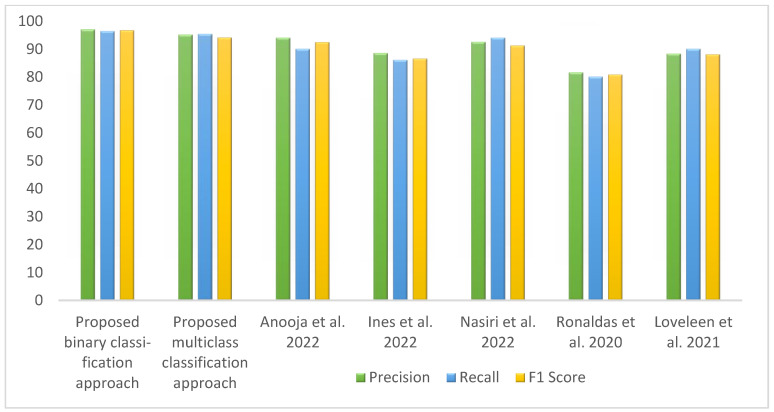
Column chart representation of the performance metrics of proposed and existing works [3,10,11,15,17].

**Figure 16 diagnostics-13-01806-f016:**
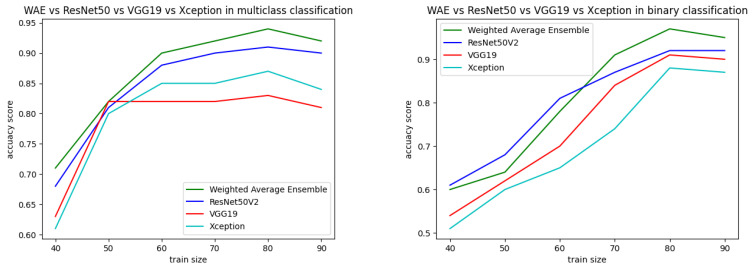
Performance analysis with different training sizes for binary and multiclass approaches.

**Figure 17 diagnostics-13-01806-f017:**
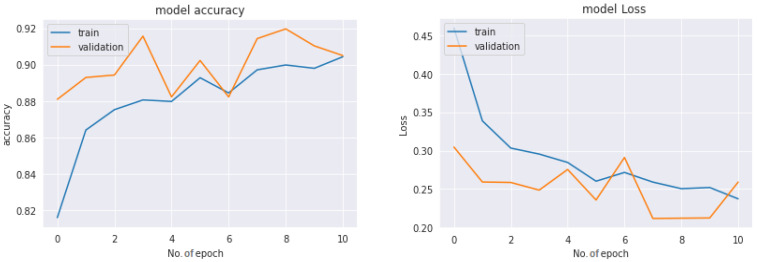
Accuracy and loss of the ResNet50V2 model-based multiclass classification system.

**Figure 18 diagnostics-13-01806-f018:**
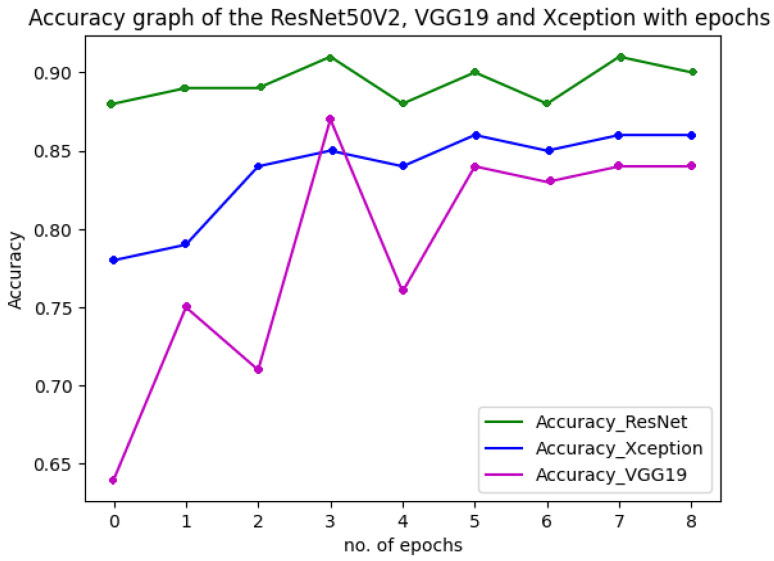
Accuracy and loss of the three models in multiclass classification system.

**Figure 19 diagnostics-13-01806-f019:**
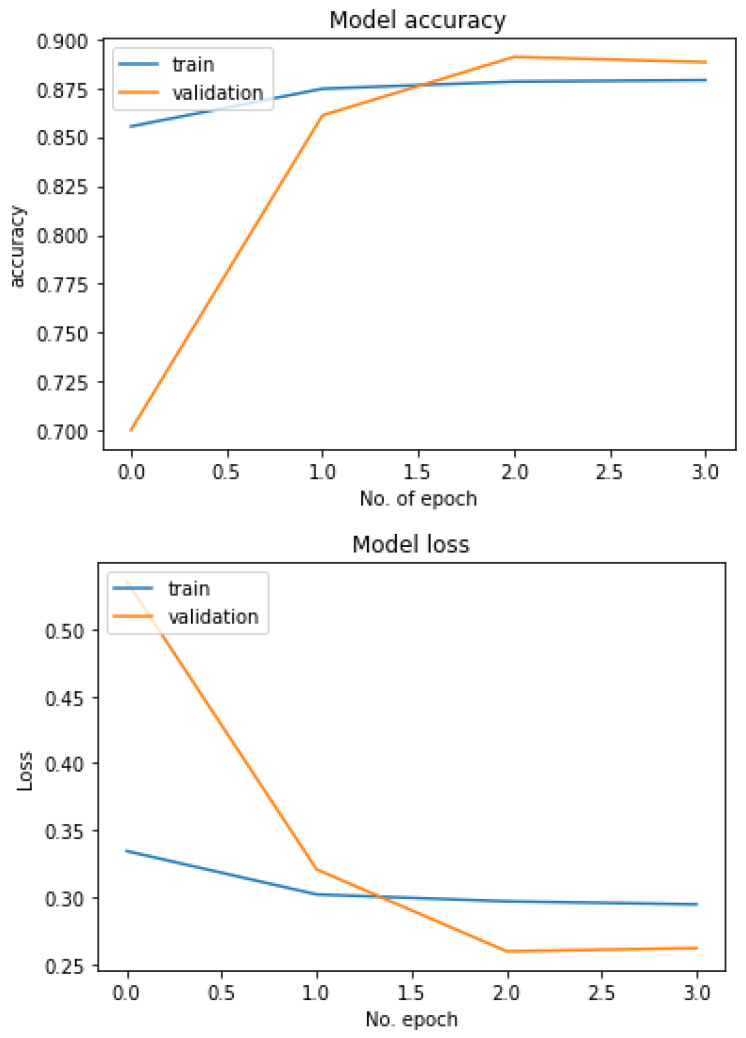
Accuracy–loss graph of the binary classification system.

**Table 1 diagnostics-13-01806-t001:** Summary of the related work.

Authors	Dataset Information	Important Remarks	Classification Type	Limitations
Areej et al. [1]	550 chest X-ray images were used that were collected from Kaggle	Binary cross entropy (BCE) loss function was used for b. classification task.	Binary classification only	Only focused on binary classification, and no proper preprocessing techniques were mentioned. Comparison of the work was also missing.
Anooja et al. [3]	Chest X-ray dataset collected from Kaggle	Nicely compared the proposed work within the existing works, and accuracy obtained from the proposed system is also high.	Only binary classification	Fewer number instances were used to train the model, which may cause reliability issues. Multiclass classification is missing. The proposed system cannot detect other diseases except COVID-19.
Wang. et al. [7]	Covidx dataset; 13,975 CXR images across 13,870 patients	Nicely introduced RT-PCR with its performances and drawbacks. CXR analysis was drawn with proper comparison.	Both binary and multiclass classification	Dataset was not updated; an updated dataset always ensures the reliability of the system.
Ines et al. [10]	Dr. Jkooy’s open-source GitHub repository, COVID-19-radiography	A detailed comparison was shown, and used both CT scan and CXR images.	Both binary and multiclass classification	Data instances were much fewer.
GJIAN et al. [13]	“http://arxiv.org/abs/2003.11597 (accessed on 11 January 2023)”	Extracted features were used as the input of SVM for classification.	Only binary classification	Data instances to train the model were much fewer, which made the reliability of the system questionable.
Ronaldas et al. [15]	“http://axiv.org/abs/2003.13865 (accessed on 18 January 2023)”	Used 3 Max-Pooling layers, 3 convolutional layers and 2 fully connected layers	Binary classification only	Only binary classification was there. Data source authenticity is questionable.
Gaur. et al. [17]	EfficientNetB0, VGG16, InceptionV3	Data instances used to train the model were properly preprocessed.	Both binary and multiclass classification	Data instances were less to make the system reliable.
Sushmithawathi et al. [19]	“https://github.com/agchung/Figure1-COVID-chestxray-dataset (accessed on 20 January 2023)”	Designed a GUI-based application that can be publicly used, developed with Google Colab GPU (Tesla K80 12GB GDDR5 VRAM).	Binary classification only	Different datasets were used separately. The ensemble technique could be used to improve the overall accuracy.
El-Kenawy et al. [24]	Two datasets were used, where the first dataset is a COVID-19 dataset with 334 CT images. The another dataset is a non-COVID-19 dataset with 794 images	Guided whale optimization technique with AlexNet model. KNN, NN, SVM classifiers were used for feature extraction.	Binary classification	The data instances used in the work were much fewer, which can create reliability issues for the proposed system.
Chang et al. [25]	DiCOVA challenge dataset 2021 was used with 1040 instances	ResNet50 model was used and ensemble technique was applied.	Acoustic-based COVID-19 classification	Dataset was not described properly, and a much lower number of instances were used.
Song et al. [26]	Two public datasets and a self-made dataset (3D-COVID) were used	U-Net and U-Net++ models were trained for segmenting the lung images.	Image segmentation	More efficient learning models can be implemented based on unlabeled data, which could reduce the scarcity of labeled data.
Sen et al. [27]	SARS-CoV-2 CT scan image database and COVID-CT database were used	Both feature extraction and feature selection techniques were developed.	Two-stage feature selection approach	Some well-established CNN models could be used to obtain better results at the initial stage.
Chang et al. [28]	DiCOVA2021 challenge dataset was used.	Both ensemble-based prediction and uncertainty estimator-based prediction were performed.	Binary classification	A proper comparison between the state-of-the-art technique and proposed technique was missing.

**Table 2 diagnostics-13-01806-t002:** Performances of the models used for multiclass classification with COVID-19 Radiography dataset.

Models for Radiography Dataset	Validation Accuracy (%)	Test Accuracy (%)
ResNet50V2 (Model3)	89.42	91.55
VGG19 (Model2)	85.50	83.04
Xception (Mode1)	87.48	87.29
Average Ensemble		93.02
Weighted Average Ensemble (WAE)		94.10

**Table 3 diagnostics-13-01806-t003:** Performances of the models used for binary classification with Covidx dataset.

Models for Covidx Dataset	Validation Accuracy (%)	Test Accuracy (%)
Xception (Model1)	85.82	88.89
ResNet50 (Model3)	89.62	92.54
VGG19 (Model2)	87.28	91.68
VGG16	71.45	73.59
Inception	81.38	83.51
Weighted Average Ensemble (WAE)		97.25

**Table 4 diagnostics-13-01806-t004:** ResNet50V2 Model summary.

Parameters	Binary Classification	Multiclass Classifiction
Val_loss	0.2619	0.2583
Val_accuracy	87.28	89.42
Training time	4 h 30 min	5 h 20 min
Epoch	4	8
Optimizer	Adam	Adam
Initial learning rate	0.001	0.001
Loss function	binary cross entropy	categorical cross entropy
Output activation fn()	Sigmoid	softmax

**Table 5 diagnostics-13-01806-t005:** Performance metrics of proposed binary and multiclass classification approaches with weighted average ensemble (WAE) technique.

	Binary Classification	Multiclass Classification
Accuracy	0.9725	0.9410
Precision	0.9695	0.9510
Recall	0.9636	0.9331
Specificity	0.9787	-
F1 Score	0.9665	0.9404

**Table 6 diagnostics-13-01806-t006:** Performance metrics of multiclass classification system with ResNet50V2.

	Precision	Recall	F1 Score
Normal	0.79232	0.92121	0.85231
Viral Pneumonia	0.99178	0.91105	0.94260
COVID	0.93418	0.86816	0.89251

**Table 7 diagnostics-13-01806-t007:** Performance metrics analysis with other DL-based existing works.

Author	Precision	Recall	F1 Score
Anooja I. et al. [3]	95	90	0.9243
Ines C. et al. [10]	88.5	86.0	0.865
H.Nasiri et al. [11]	92.5	95	0.912
Ronaldas M.J. et al. [15]	81.57	80.07	0.8081
Loveleen G. et al. [17]	88.3	90.0	0.88
Proposed binary classification technique with WAE	96.95	96.36	0.9665
Proposed multiclass classification technique with WAE	95.1	95.31	0.9404

**Table 8 diagnostics-13-01806-t008:** Comparative analysis between the proposed work and state-of-the-art techniques.

Author	Models and Methods	Total Images	Accuracy
Areej et al. [1]	Transfer learning with pretrained ChexNet model	560	89.8273
Anooja et al. [3]	AlexNet	7240	93.65
Ines et al. [10]	VGG-19, Xception	2022	90.496
CHANGJIAN et al. [13]	Transfer learning on ResNet	1495	93
Ronaldas et al. [15]	CNN with Lim. Adaptive histogram equalization technique	702	83.29
Loveleen et al. [17]	EfficientNetB0, VGG16, InceptionV3	5010	92.89
Sushmithawathi et al. [19]	DenseNet201, ResNet50V2, Inceptionv3	7240	91.31
El-Kenawy et al. [24]	Guided whale optimization technique with AlexNet, KNN, NN, SVM	1128	79.0
Chang et al. [25]	Ensemble technique with ResNet50	1040	84.53
Song et al. [26]	Image segmentation with U-Net and U-Net++ models	2922	95.49
Sen et al. [27]	CNN models, Dragonfly feature selectioin technique with SVM classifier	2482	90.00
Chang et al. [28]	Gaussian noise-based augmentation technique and transfer learning with pretrained ResNet50 model	1040	91
He et al. [38]	Image segmentation with evolvable adversarial network using gradient penalty	1199	94
Proposed binary classification approach	Weighted average ensemble technique with ResNet50V2, VGG19 and Xception	30,386	97.25
Proposed multiclass classification approach	Weighted average ensemble technique with VGG19, Xception, ResNet50V2	7461	94.10

**Table 9 diagnostics-13-01806-t009:** Performance analysis with different train–test sizes.

Train–Test Ratio in Multiclass Approach	Accuracy with Weighted Average Ensemble	Accuracy with ResNet50v2 Model	Accuracy with VGG19 Model	Accuracy with Xception Model
90:10	92.94	90.58	81.04	84.13
80:20	94.10	91.55	83.04	87.29
70:30	92.65	90.80	82.98	85.12
60:40	90.16	88.85	82.65	85.03
50:50	82.01	81.87	82.02	80.48
40:60	71.47	68.23	63.42	61.53
**Train–Test Ratio in Binary Approach**	**Accuracy with Weighted Average Ensemble**	**Accuracy with ResNet50v2 Model**	**Accuracy with VGG19 Model**	**Accuracy with Xception Model**
90:10	95.29	92.33	90.08	87.89
80:20	97.25	92.54	91.68	88.89
70:30	91.65	87.65	84.93	74.65
60:40	78.21	81.35	70.64	65.53
50:50	64.30	68.49	62.67	60.68
40:60	60.01	61.64	54.52	51.67

**Table 10 diagnostics-13-01806-t010:** Performance analysis with 80:20 train–test ratio.

Experiment No.	Multiclass Classification Accuracy	Binary Classification Accuracy
Learning experiment 1	94.10	97.25
Learning experiment 2	94.09	97.21
Learning experiment 3	94.12	97.24
Learning experiment 4	94.10	97.25
Learning experiment 5	94.10	97.25
Learning experiment 6	94.08	97.24
Learning experiment 7	94.09	97.24
Learning experiment 8	94.08	97.24
Learning experiment 9	94.10	97.25
Learning experiment 10	94.10	97.24
Mean accuracy	94.096	97.24
Standard deviation	0.0117	0.0119

**Table 11 diagnostics-13-01806-t011:** Performance analysis with 5-fold cross-validation method.

Split Count	WAE in Binary Approach	WAE in Multiclass Approach
1	87.24	90.41
2	81.67	81.35
3	91.74	88.64
4	79.61	84.32
5	90.25	80.56
Mean	86.1	86.05
Standard deviation	5.29	4.98

## Data Availability

Datasets used in this work are publicly available.

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
