# Peer review of "A Novel Deep Learning-Based Classification Framework for COVID-19 Assisted with Weighted Average Ensemble Modeling"

_diagnostics, 2023, doi:10.3390/diagnostics13101806_

Round 1

Reviewer 1 Report

This manuscript looks like good material. But it must be "cleaned" and mature a bit more. My major comments are as follows:

1) Please revise the Abstract by presenting your motivation and your main contributions to this manuscript. You should propose your main work before lines 3 or 4. So, please summarize the overall introduction in the Abstract (instead you can mention them in the Introduction).

2) Please mention a brief detail about the dataset and provide your main findings (accuracy, precision, recall, etc., or the improvement rate by ensemble model).

3) Please provide your main contributions clearly at the end of the Introduction. I suggest using a bullet list point. What are your main idea and contributions against many existing  homogeneous and weighted heterogeneous ensemble models?

4) Review of the literature should be effectively enhanced. Please find and study/compare (from a broader viewpoint) the works of Zeinab Ghasemi Darehnaei, Nazanin Pilevari, Seyedali Mirjalili, Maryam AlJame, Mahtab Ranjbarimesan, and Shaoze Cui, on Ensemble Learning, Covid-19 related works, and Deep Learning. 

5) The weighted average ensemble for classification tasks is better when we utilize much more base learners, not only three ones (as in your paper). Please clarify it. Moreover, how you specified these weights? What about the final thresholding parameter to design the final output of the model?  Please provide more details about calculating the final output of the ensemble mode (Figure 7).

7) Discuss the parameter setting of the different deep learning models? The values of the parameters could be a complicated problem itself, So how do the authors give the values of these parameters?

8) Look for a broader view in the Conclusion and Outlook section. Find and see further works by other teams as mentioned in the fourth comment. For example, the hyperparameters of the ensemble model can be optimized via a metaheuristic algorithm. Moreover, to make the ensemble learning model be more accurate and useful, you can also use different classifiers (MLP, KNN, SVM, Decision Trees, …) as the classifier of each deep learning model (to provide a two-level ensemble model). I suggest mentioning these techniques as well as other suggestions as potential future research directions to further improve your work in the future. You are suggested to use proper references for each suggestion at the End of the Conclusion. If it could be done, it can become a fantastic paper.

Please double-check the whole manuscript to be free of Grammatical errors.

Reviewer 2 Report

The paper developed covid-19 classification using weighted Average Ensemble Modeling.  Three deep learning models were combined aiming at improving the accuracy of the previous COVID-19 classification algorithms. The paper needs major revision before it is accepted for publication.

Specific comments are as follows:

1.      There is not too much novelty in the paper. Please point them out and use a bullet list to list them at the end of the introduction.

2.      There have been many better classification techniques authors for covid-19 classification.

The authors are expected to provide a comparison of experimental results with these new algorithms. If the authors cannot provide comparison experimental results, at least they need to cite and compare the methods using languages.  The techniques that need to be cited, and compared are:

·        “A COVID-19 Detection Algorithm Using Deep Features and Discrete Social Learning Particle Swarm Optimization for Edge Computing Device”, ACM Transaction on Internet Technology, Vol. 22, No. 3, 2022.

·        “COVID-19 Infection Segmentation and Severity Assessment Using a Self-Supervised Learning Approach” Diagnostics, vol. 12, no. 8: 1805,2022.

·        "An evolvable adversarial network with gradient penalty for COVID-19 infection segmentation", Volume 113, Part B,107947, Applied Soft Computing, December 2021.

3.      The authors use a table to show the past work. Could you please organize them based on the methods the papers use?

4.      The description of the datasets is not clear. You need to provide the size, how to acquire the images, etc.

5.      Please provide the training time and convergence curve with iterations for the training of the model.

6.      How to weigh the three CNN models is not clear.

7.      Please provide comments on how to combine with non-CNN model.

Please check carefully.

Round 2

Reviewer 1 Report

Thank you for accurately revising the manuscript. Now, it can be accepted.

Author Response

Thanks for the kind response. Your suggestions in round 1 were very constructive and they helped a lot to improve our paper.